# The Effect of the Cordillera Mountain Range on Tropical Cyclone Rainfall in the Northern Philippines

**Bernard Alan B. Racoma** [1,2,*] **, Christopher E. Holloway** [1] **, Reinhard K. H. Schiemann** [3] **, Xiangbo Feng** [3] **and Gerry Bagtasa** [2]

1   Department of Meteorology, University of Reading, Reading RG6 6BB, UK
2   Institute of Environmental Science and Meteorology, University of the Philippines, Diliman, Quezon City 1101, Philippines
3   National Centre for Atmospheric Science, Department of Meteorology, University of Reading, Reading RG6 6BB, UK
*   Correspondence: b.b.racoma@pgr.reading.ac.uk

**Abstract:** In this study, we examined the sensitivity of tropical cyclone (TC) characteristics and precipitation to the Cordillera Mountain Range (CMR) in Luzon, Philippines. Using the Weather Research and Forecasting (WRF) model, we simulated eight TCs with four different CMR orographic elevations: Control, Flat, Reduced, and Enhanced. We found that at significance level $\alpha = 0.05$, TC intensity significantly weakened as early as 21 h prior to landfall in the Enhanced experiment relative to the Control, whereas there was little change in the Flat and Reduced experiments. However, throughout the period when the TC crossed Luzon, we found no significant differences for TC movement speed and position in the different orographic elevations. When a TC made landfall, associated precipitation over the CMR increased as the mountain height increased. We further investigated the underpinning processes relevant to the effect of the CMR on precipitation by examining the effects of mountain slope, incoming perpendicular wind speed, and the moist Froude Number ($F_w$). Compared with other factors, TC precipitation was most strongly correlated with the strength of approaching winds multiplied by the mountain slope, i.e., stronger winds blowing up steeper mountain slopes caused higher amounts of precipitation. We also found that individually, $F_w$, mountain height, and upslope wind speeds were poor indicators of orographically induced precipitation. The effects of mountain range on TC rainfall varied with TC cases, highlighting the complexity of the mountain, wind, and rainfall relationship.

**Keywords:** tropical cyclones; precipitation; orographic effect; numerical weather prediction; Philippines

## 1. Introduction

Land elevation has been shown to have a large influence on local rainfall, a phenomenon known as orographic precipitation [1–3]. The forced mechanical lifting of moist air along slopes leads to condensation and then precipitation along the windward flanks of the mountain [3]. Orographic precipitation has been widely studied; however, there is limited literature focusing on the orographic effect during tropical cyclones (TCs), especially in the Philippines.

Racoma et al. [4] found that the distribution of TC rainfall in the Philippines is sensitive to the TC track. Different TC tracks have different windward sides where precipitation could be orographically enhanced [4]. Using observations, another study by Racoma et al. [5] confirmed that stronger TCs tend to produce more precipitation along the mountainous regions of Luzon. They investigated the overall statistical relationships between TC rainfall and TC characteristics (cyclogenesis, track, and intensity), and other environmental factors such as environmental moisture and seasonality. However, it remains unclear what processes determine the TC precipitation patterns over land.

Diagnostic models for orographic precipitation have been proposed in previous studies. Sinclair [6] devised a simple model estimating orographic precipitation by determining orographically induced vertical motion, where condensation was assumed to arise due to saturated ascent. In a more recent study, Rostom and Lin [7] devised a modified Orographic Rain Index (ROI) as a predictor for TC precipitation in the Appalachian Mountains. The ROI considers the following factors: incoming horizontal wind speed, slope of the mountain, relative humidity, TC translation speed, and TC size [7]. Although topography may suppress precipitation due to factors such as forced descent, flow blocking, vertical mixing, and the depletion of atmospheric moisture, precipitation along the windward side and mountain top may be enhanced due to convection caused by terrain-induced ascent overcoming local convective inhibition [8]. Using idealised simulations, Lin et al. [9] reported that landfall location and approach angle over mountain ranges may affect TC movement. This has profound effects on the spatial distribution of precipitation [9]. Similarly, in a series of numerical experiments, Liu et al. [10] determined that when passing over mountains, idealised TCs are deflected to the south upstream, continue over the mountain anticyclonically, and then resume moving west.

Some previous studies of orographic precipitation of TCs in Asia have focused on the effects of Taiwan's Central Mountain Range on TCs. Using radar and rain gauge observations, Yu and Cheng [11] revealed that in Typhoon Morakot (2009), rainfall in the southwestern region of the Central Mountain Range was orographically enhanced and was proportional to the oncoming wind speed multiplied by the background precipitation. Additionally, Agyakwah and Lin [12] found that the extreme precipitation during Typhoon Morakot (2009) was due to a combination of four key processes: the initialisation of rainfall by the southwest monsoonal current; moving in off the TC rainbands during TC approach; merging of the southwest monsoonal currents and the rainbands; and finally, the merging of orographic rain and TC rain, leading to convective enhancement. Furthermore, idealised studies have revealed that higher wind speeds from TCs tend to produce higher amounts of rainfall towards the southern region of the Central Mountain Range in Taiwan [13]. A study by Tang and Chan [14] used the Weather Research and Forecasting (WRF) model, a numerical weather prediction (NWP) model to simulate the effects of the Central Mountain Range of Taiwan and the mountain ranges of Luzon, Philippines, on incoming TCs. They found that in Taiwan, TC tracks are deflected towards the north prior to landfall due to the asymmetric diabatic heating caused by the topography of the Central Mountain Range. They further argued that the mechanisms that occur in the Central Mountain Range may not be the same for the Philippine mountain ranges due to the lower mountain height and moister air over the mountains on Luzon Island.

In this study, we aimed to use the WRF model to investigate how the orography of Luzon, particularly the Cordillera Mountain Range (CMR), affected TC-associated precipitation. Additionally, we evaluated how the CMR affects characteristics such as movement speed, intensity, and wind strength, and how these characteristics are related to TC precipitation. The CMR induces monsoon blocking effects affecting precipitation over Luzon [15]; TCs contribute as much as 54% of precipitation along the western coast of Luzon [16]. Additionally, two major river basins of the Philippines are located within the boundaries of the CMR, with the Cagayan River basin to the east and the Pampanga River basin to the south of the mountain range. Both river basins have experienced major flooding in recent years [17,18], and understanding how the CMR affects TC precipitation, especially for stronger TCs, may be helpful in regional flood forecasting efforts.

This paper is organised as follows: In Section 2, we begin with the Data and Methodology section, where we discuss the data and the model used for this study. In this section, we also describe the sensitivity experiment setup in which we modified the topography of Luzon. Section 3 presents model outputs and comparisons between the different model runs. Finally, we summarise the paper in the final chapter, where we also discuss the caveats of the present study, as well as potential future research.

## 2. Data and Methodology

### 2.1. Historical TC and Precipitation Data

To evaluate the model outputs for this study, we used historical best-track TC data from the International Best Track Archive for Climate Stewardship v04r00 (IBTrACS) and precipitation data from Integrated Multi-satellitE Retrievals for GPM (IMERG). IBTrACS is a publicly available centralised repository of historical best-track TC data, where records of TC intensity, track, and movement speed of different agencies are stored in a central database [19,20]. The database includes 6-hourly TC tracks from as early as the 1800s, although TC wind and intensity data are only available from 1978 onwards, with most 3-hourly records made available via interpolation [4,19,20]. On the other hand, IMERG provides spatial rainfall by combining information from multiple satellites [21,22].

### 2.2. Weather Research and Forecast Modelling

We used the WRF model version 3 [23] to investigate the effects of the CMR located in Luzon, Philippines, on the precipitation and characteristics of selected TCs. A summary of the options used is listed in Table 1. Choices of physical parameterisation schemes and model configuration were adapted from the operational forecasting configurations of the Philippine weather bureau, Philippine Atmospheric, Geophysical and Astronomical Services Administration [24], optimisation studies by Tolentino and Bagtasa (2021) [25], historical comparisons by Bagtasa (2021) [26], and sensitivity tests for TC simulations conducted by Delfino et al. (2022) [27]. Additionally, we adapted the Planetary Boundary Layer scheme based on the study of Cruz and Narisma [28] due to its performance in simulating heavy precipitation over Luzon. The European Centre for Medium-Range Weather Forecasts Reanalysis fifth-generation (ERA5) [29] data are used as the initial and boundary conditions for the WRF model. We defined two nested domains with horizontal grid spacings of 15 km and 5 km (Figure 1). The outer domain was configured to resolve the genesis and development of a TC over the Pacific Ocean, while the inner domain was configured to simulate precipitation over the Philippines. In the WRF model output, explicitly resolved non-convective rainfall (RAINNC) was recorded separately from convective rainfall (RAINC) generated from the selected cumulus parametrisation scheme [23]. For the inner domain, although we used the Kain–Fritsch convection scheme, we note that 70.09% of the rainfall over Luzon and 75.63% of the rainfall over the CMR region is generated by RAINNC, i.e., the large-scale explicit rainfall (when averaged for all models 24 h from landfall). This gives us confidence that we are studying physically based convective rainfall processes. For the model runs, the "adaptive time-step" option of WRF was enabled for model stability and efficiency [23], and the model output was set at 3-hourly intervals. To achieve a balance between longer simulation periods to allow for spin up, TC development and maturation [27] and shorter simulation periods for higher precipitation accuracy [30], simulations were initialised at 96 h before TC landfall and ended at 24 h after landfall. This allowed for a simulation period of 5 days: 48 h spin up time, 48 h pre-landfall TC development, and 24 h post-landfall analysis.

We simulated eight TCs that made landfall in Luzon between 2000 and 2015. The landfall dates and landfall intensities of these TCs are listed in Table 2. These TCs caused extreme precipitation based on the criteria defined by Racoma et al. [5], where extreme precipitation is the area mean of precipitation over Luzon exceeding the local 95th percentile precipitation threshold. Additionally, in the initial model simulations, these TCs intensified along the Pacific Ocean, tracked from east to west, made landfall along Luzon, and then crossed the CMR in a westerly movement.

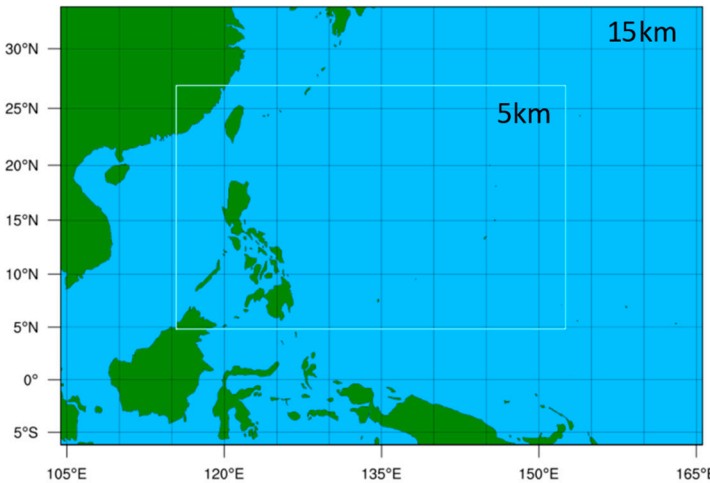

**Figure 1.** The two-way nested model domains, with the outer domain at 15 km resolution, and the inner domain at 5 km.

**Table 1.** Summary of selected WRF configuration schemes.

| Number of Domains | Two |
|---|---|
| Horizontal Grid Spacing | 15 km (outer), 5 km (inner) |
| Cumulus Parameterisation | Kain–Fritsch Scheme |
| Microphysics | WRF Single-Moment 6-class Microphysics Scheme |
| Planetary Boundary Layer | ACM2 Scheme |
| Longwave Radiation | Rapid Radiative Transfer Model Scheme |
| Shortwave Radiation | Dudhia Scheme |
| Surface Physics | Unified Noah Land-Surface Model |
| Model Time-Step | Adaptive Time-Step |
| Initial and Boundary Conditions | ERA5 |

**Table 2.** The eight TCs simulated in this study. Their landfall dates are listed in the second column; the intensity upon Luzon landfall is presented in the third column; and the direct positional error (DPE) of each TC is detailed in the fourth column. The calculated DPE has a standard deviation of 36.70 km, mean of 118.31 km, and median of 121.86 km.

| TC Name | Date of Philippine Landfall | Intensity Upon Luzon Landfall (Millibars) | DPE (km) |
|---|---|---|---|
| Bebinca | 2 November 2000 | 990 | 128.00 |
| Imbudo | 22 July 2003 | 950 | 115.73 |
| Prapiroon | 31 July 2006 | 1002 | 142.06 |
| Cimaron | 29 October 2006 | 920 | 102.06 |
| Megi | 18 October 2010 | 885 | 182.56 |
| Nesat | 26 September 2011 | 950 | 66.67 |
| Nari | 11 October 2013 | 970 | 141.32 |
| Koppu | 17 October 2015 | 925 | 68.07 |

We then modified the orography of the CMR (the main mountain range within the red region enclosed in Figure 2) in the sensitivity experiments of WRF simulations. In addition to the original orography (Control, Figure 2a), the elevation of the CMR was set to zero in the Flat experiment (Figure 2b), reduced by a factor of two in the Reduced experiment (Figure 2c), and enhanced by a factor of two in the Enhanced experiment (Figure 2d).

The edges of the modified orography were also smoothed along the perimeter so as not to introduce large deviations between the modified CMR and its neighbouring regions. Modifying the orography enabled investigation of the interactions between the winds of TCs and CMR, and their effects on precipitation and TC characteristics. The differences in the TC characteristics and precipitation for different simulations were then evaluated at a significance level $\alpha = 0.05$. For each 3-hourly interval, we calculated the $100 \times (1 - \alpha)\%$ or the 95% confidence interval (CI) based on the standard error (SE) by adding or subtracting $1.96 \times$ SE from the mean.

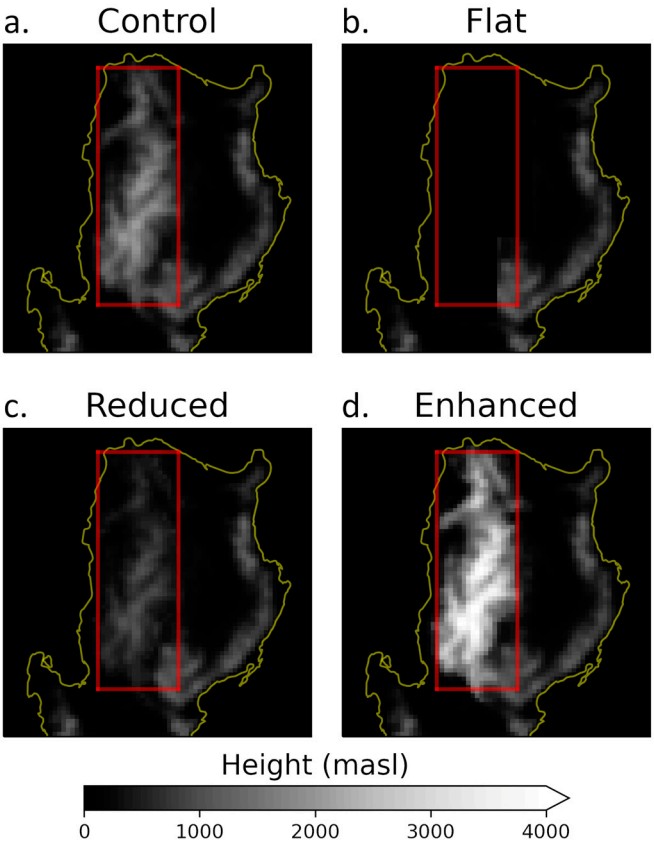

**Figure 2.** The orographic height in metres above mean sea level (masl) for (**a**) Control, (**b**) Flat, and (**c**) Reduced, and (**d**) Enhanced orography experiments. The CMR is enclosed in the red box.

## 3. Results

### 3.1. Evaluation of Control Simulations against Observations

We first plotted the TC tracks derived from the Control simulations and compared them with tracks from IBTrACS (black tracks in Figure 3). To evaluate the modelled TC track accuracy, we calculated the direct positional error (DPE) [31] between IBTrACS and the TC tracks from the Control simulations. The DPE is defined as the geodesic distance between TC position in the model simulations and the best track [32]. For the DPE, we used the 3-hourly TC track points from 48 h before landfall until 24 h after TC landfall.

From Table 2, the DPE of the TCs in the Control runs ranged from a minimum of 66.67 km (Nesat) to a maximum of 182.56 km (Megi). The DPE for the eight TCs had a standard deviation of 36.70 km, a median of 121.86 km, and an average of 118.31 km. For the adapted parametrisation options, the aforementioned values show that most TC tracks are reasonably modelled compared with observations, in the sense that we can analyse these TCs to answer the research questions of this study. Although the modelled TC tracks did not perfectly coincide with the observations, the simulated TCs still moved along Luzon and across the CMR during each of the model runs.

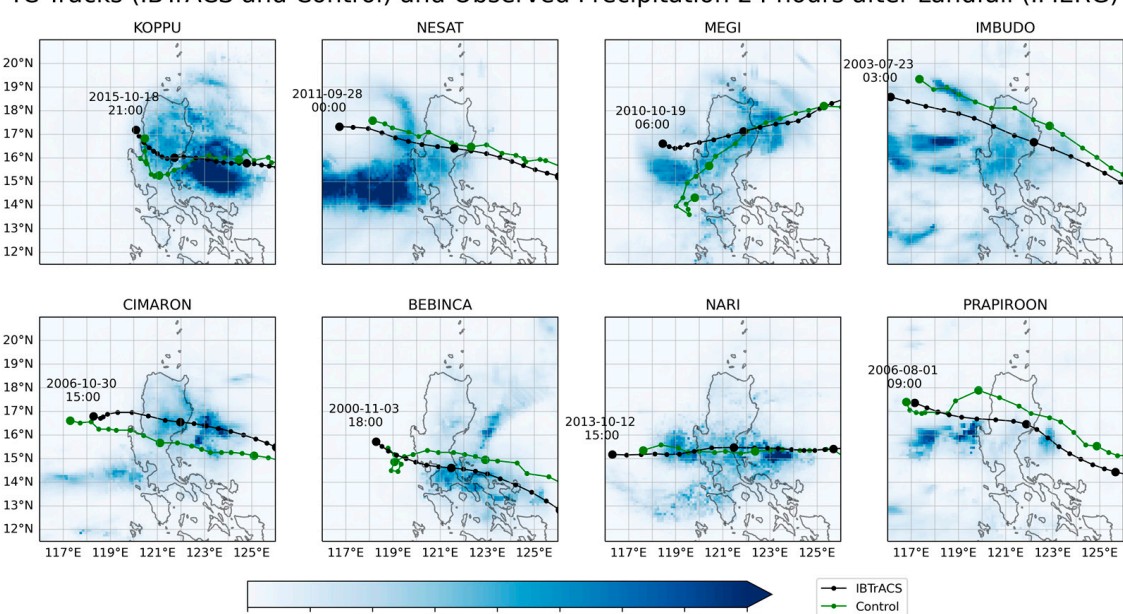

**Figure 3.** Three-hourly observed (black) and simulated (green) TC track points for Control for all eight cases. These are plotted for up to 24 h after observed TC landfall based on IBTrACS. Larger markers denote 24 h periods, matched between observations and simulations, while the final point in the model track corresponds to the same time and date as the labelled observed point. The blue shading denotes the 24 h accumulated precipitation (in mm) from IMERG. Observed track data are from IBTrACS [19], while precipitation data are from IMERG [21].

The 24 h post-landfall accumulated precipitation of the observations and Control are shown in Figures 3 and 4, respectively. The modelled precipitation differed significantly from the observed IMERG precipitation (Figure 3), with the Control (Figure 4) showing precipitation over the mountain ranges of Luzon that were not detected by IMERG. Although IMERG does accurately measure regional precipitation patterns, it has large uncertainty in measuring precipitation over mountainous terrain [33–35]. Although uncertainties remain in the results of the WRF model, detailed evaluation of model accuracy, as well as its improvement, is not within the scope of this paper. We aimed to understand the sensitivity of TCs on mountain ranges; therefore, the WRF model, which can produce the mountain-associated features in rainfall distribution, allowed us to quantitatively measure precipitation over Luzon and over the CMR. As such, we believe that the WRF model is sufficient for the aims of this study.

To understand how the CMR affects TC precipitation, we analysed the accumulated precipitation 24 h after TC landfall for the Control simulation. For most of the TC simulations, heavy precipitation was found along the eastern slopes of the CMR (Figure 4). However, for Imbudo and Prapiroon, there was more precipitation towards the western section of the CMR. This difference in precipitation distribution can be attributed to the different windward and leeward sides based on the TC cyclonic flow [4]. To consider the sensitivity of precipitation to TC track with respect to the mountain range, hereafter we classify the TCs according to two categories depending on the movement and wind direction of TC circulation across the CMR: the north-tracking TCs (NTCs: Imbudo and Prapiroon) with mostly westerly winds over CMR; and south-tracking TCs (STCs: the other six TCs) having mostly easterly winds.

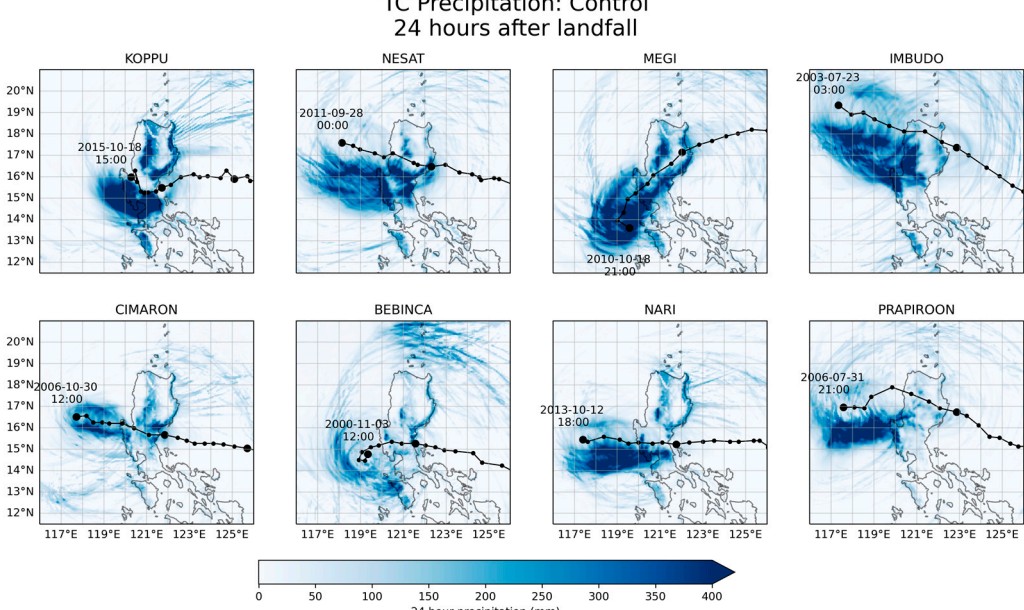

**Figure 4.** Accumulated precipitation 24 h after simulated landfall (shading) for Control, with the simulated 3-hourly TC tracks in black lines. Tracks are plotted up to 24 h after landfall, with each 24 h period plotted in larger markers.

### *3.2. Effect of Mountain Height on TC Intensity and Track*

To understand the effects of mountain height on TC characteristics, we evaluated the differences in TC intensity and movement between the modified orography experiments (Flat, Reduced, and Enhanced) and the Control simulation from 48 h before landfall up to 24 h after landfall. We first calculated the mean intensity, latitude, and movement speed of the TCs for the modified orography experiments and then compared them with the Control by taking the differences between the experiments. These differences at 3-hourly intervals are plotted in time-series in Figure 5, where the mean differences for all cases are plotted in solid black lines, and the 95% CIs based on the standard SE are plotted in grey shading. The 95% CIs were obtained by adding or subtracting 1.96 × SE from the means.

Both the mean difference and SE were close to zero during most of the 72 h period; therefore, both the Flat and Reduced experiments showed no significant effect on TC intensity (Figure 5a,b). However, increasing the height of the CMR in the Enhanced experiment significantly weakened most TCs as early as 21 h prior to landfall (Figure 5c). It is expected that TCs weaken as they make landfall, especially if they traverse mountain ranges [10,13,36]. However, no significant change in TC strength was observed in the Flat and Reduced experiments, despite the flatter terrain compared with the Control. For the eight TCs simulated in this study, TC intensity was sensitive to a higher CMR, but not sensitive to lower CMR heights. This indicated nonlinear relationships between CMR height and TC metrics. Additionally, this means that the actual CMR appeared to have minimal effects on the intensity of passing TCs (as least for these case studies) because its influence did not significantly differ between the Control, Flat, and Reduced. We also found that changing the mountain height had little to no significant effect on TC meridional position (Figure 5d–f) and movement speed (Figure 5g–i). In terms of track, we found no systematic changes in TC track with respect to varying CMR heights (Figure 6).

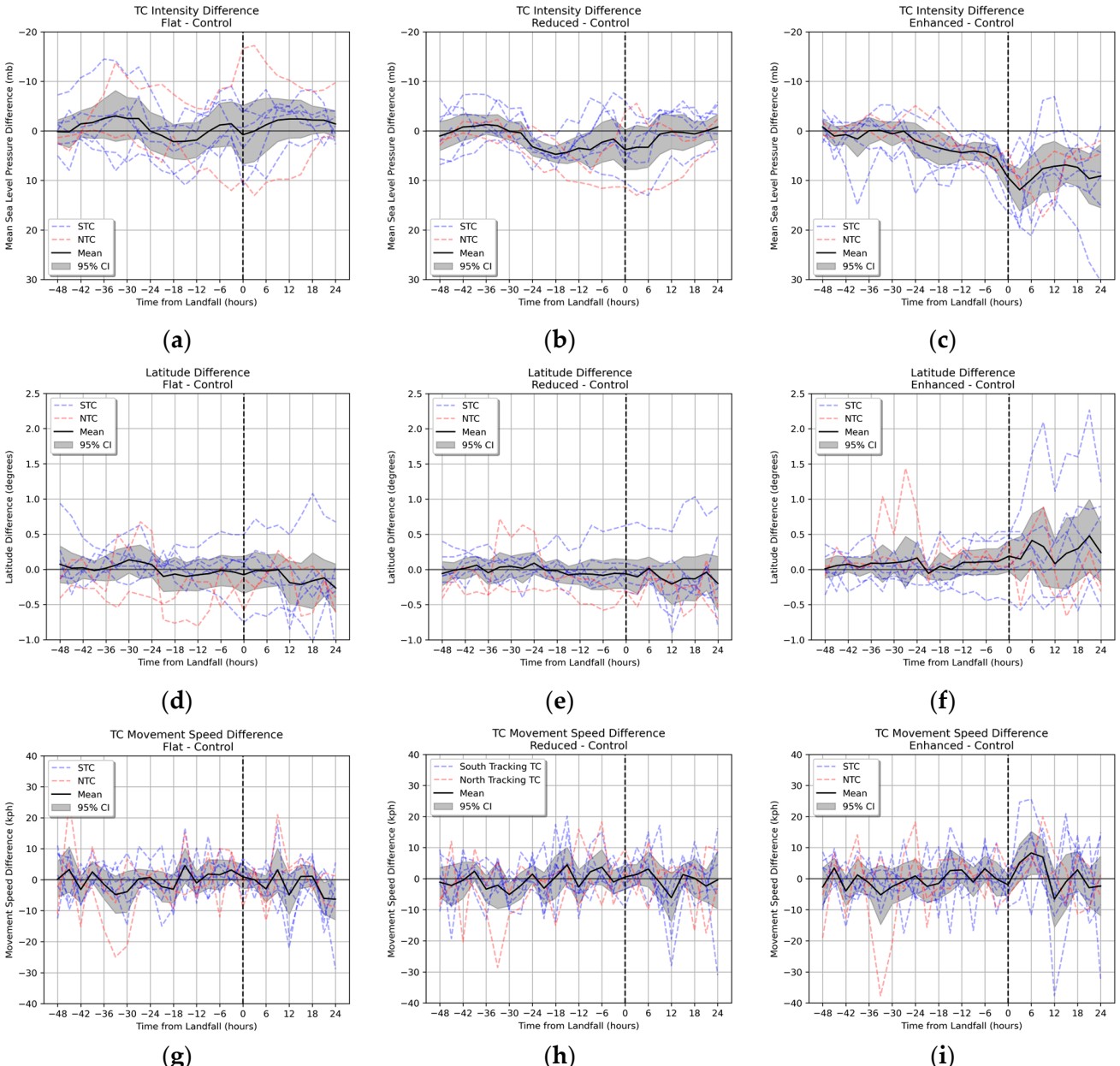

**Figure 5.** TC intensity comparison for (**a**) Flat—Control, (**b**) Reduced—Control, and (**c**) Enhanced—Control; TC location comparison for (**d**) Flat—Control, (**e**) Reduced—Control, and (**f**) Enhanced—Control; TC movement speed comparison for (**g**) Flat—Control, (**h**) Reduced—Control, and (**i**) Enhanced—Control. Take note of the inverted y-axis for (**a**–**c**) to highlight the decrease in the intensity of the TCs. The red dashed lines represent the NTCs, with the blue dashed lines representing the STCs. The black lines represent the means of all cases, with the grey shading showing the 95% CIs based on SE by adding or subtracting 1.96 × SE from the means. The vertical black dashed lines are the time of landfall, or $t = 0$.

**Figure 6.** TC tracks for the different orography experiments: Flat (black), Reduced (blue), Control (magenta), and Enhanced (red). Tracks are plotted up to 24 h after landfall, with each 24 h period plotted in larger markers.

### 3.3. Effect of Mountain Height on TC Precipitation

We next compared maps of the differences in 24 h post-landfall accumulated precipitation between the orography experiments. Notably, there was less rain along the eastern slopes of the CMR in all Flat experiments (Figure 7) except for Imbudo, wherein increased rainfall was seen in the immediate vicinity of its track. A closer look at the 24 h precipitation difference between the Flat and Control is shown in Figure S1 in the Supplementary Materials. In the Reduced experiments (Figure 8, with a zoomed-in version shown in Figure S2), the spatial distribution of the dryer regions seen in the Flat experiments are still apparent, but less abundant. In contrast, the Enhanced experiments (Figure 9, with a zoomed-in version shown in Figure S3) show a shift to a wet difference on the eastern slopes of the CMR except for the two NTCs—Imbudo and Prapiroon. The dry regions across the CMR for the two NTCs is explained by two reasons: one is the reduction in the TC intensity in the Enhanced runs, as discussed above, with the reduced overall rain-rate of the TCs; secondly, the windward side of the CMR shifted to the west of the mountain range for the two NTCs, with the higher mountain height thus serving to block the rainfall induced by the TC circulation along the CMR, resulting in less rain in the cases of Imbudo and Prapiroon. The results show significant changes in rainfall amounts for varying CMR heights, which confirms that TC–orography interactions play a major role in TC rainfall production along the CMR region.

A closer inspection of the simulated TC tracks revealed that there was a northward shift in the tracks of Bebinca and Nari with respect to their respective Control runs, particularly for the Enhanced and Reduced (for Bebinca only) simulations. The track shifts alone clearly contributed to the rainfall differences notwithstanding the changes in orography. We distinguished the precipitation difference from the track shifts and orographic height changes by first determining the radial TC–rainfall profile from the Control model output along the eastern slopes of the CMR where the differences are most prominent. We found that the Control TC–rainfall profile of Nari decays logarithmically the farther the distance from the TC centre, whereas the Control rainfall of Bebinca follows an almost-flat profile that gradually tapers at distances > 250~300 km from the TC centre. The difference in the rainfall profiles is due to the difference in landfall intensities where Nari made landfall as a typhoon and Bebinca only materialised as a tropical storm (their intensities are shown in Table 1). We estimated the mean track shifts of 32 km and 51 km for Nari and Bebinca,

respectively. We then shifted the Control rainfall profiles of both TCs according to their respective track shifts to calculate the rainfall differences due to their tracks shifting in the Enhanced runs. In the Enhanced runs, we found that for Nari, approximately 75% of rainfall changes were due to the orography change and 25% were due to its track shifting 30 km northward, closer to the CMR. In the case of Bebinca, approximately 90% of the rainfall difference was attributed to the orographic change and approximately 10% was due to the TC moving 50 km closer to CMR. Thus, we conclude that the effect of track shift (which is mostly minor in these six cases) on TC rainfall changes is expected to be small.

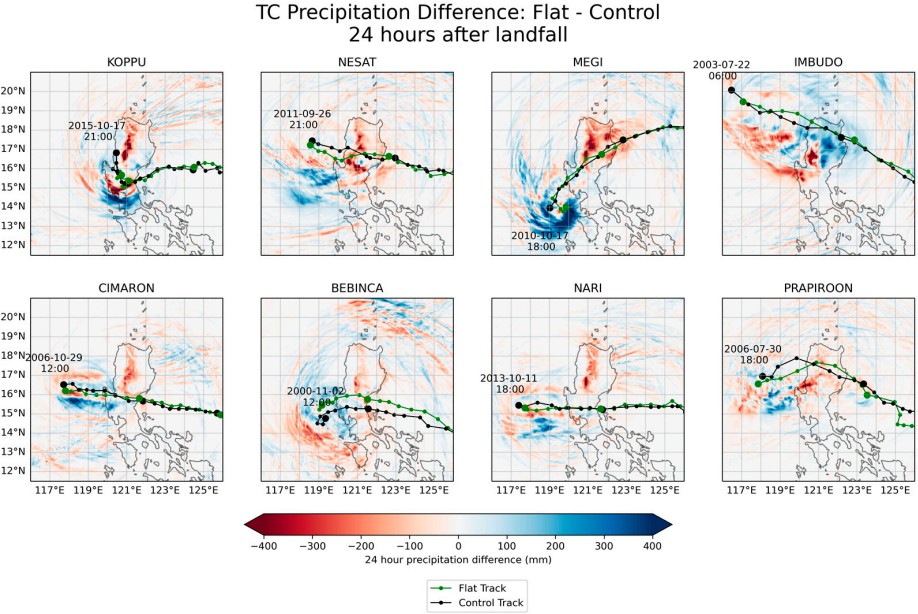

**Figure 7.** Difference in 24 h accumulated precipitation after landfall (shading) between Flat and Control. Three-hourly TC tracks for Flat and Control are coloured in green and black, respectively. Tracks are plotted up to 24 h after landfall, with each 24 h period plotted in larger markers, with the final points in both model tracks corresponding to the same time and date as labelled.

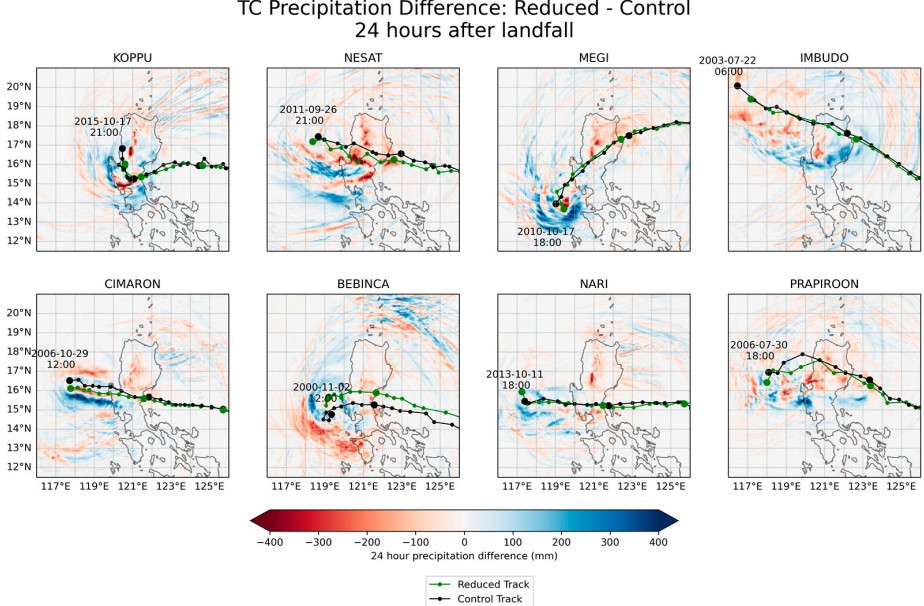

**Figure 8.** Similar to Figure 7, but for the difference in 24 h accumulated precipitation after landfall between Reduced and Control.

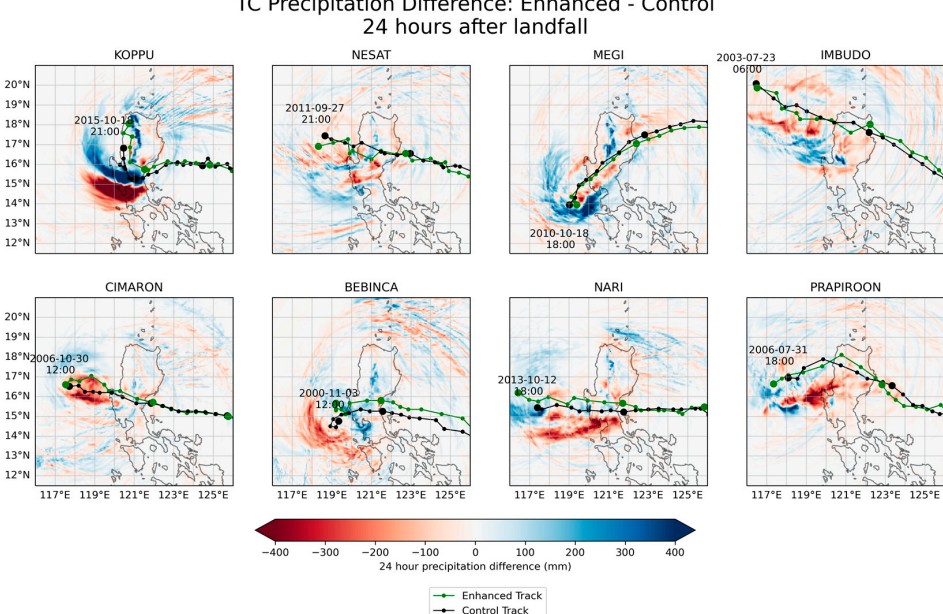

**Figure 9.** Similar to Figure 7, but for the difference in 24 h accumulated precipitation after landfall between Enhanced and Control.

We then evaluated the differences in the accumulated precipitation between the experiments and Control at $\alpha = 0.05$. This was performed by resampling the 5 km grids to a coarser 25 km grid resolution, conducting paired t-tests for the accumulated precipitation of the TCs for the different experiments, and then mapping regions where the *p*-value was less than 0.05 (Figure S4 in the Supplementary Materials). For both the Flat (Figure S4a) and Reduced (Figure S4b) experiments, reducing the height of the CMR had a significant effect on the precipitation over the mountain range. The change in precipitation in the Enhanced experiment was not significant over Luzon when all eight TCs were considered (not shown). However, when only the STCs were included, at $\alpha = 0.05$, the CMR had a significant effect on precipitation over the mountain range (Figure S4c).

Differences in the amount of TC precipitation due to the change in mountain height showed regional variations, it is also necessary to estimate the precipitation response as the TC moves along Luzon. For every 3-hourly model output timestep from 48 h prior to ($t = -48$) and until 24 h after landfall ($t = 24$), we calculated the mean overland precipitation over Luzon and over the CMR. For the mean overland precipitation over Luzon, we found that there was no statistically significant difference for Flat, Reduced, and Enhanced when compared with Control (not shown). Although there were indeed different spatial patterns in overland precipitation in the different orography experiments (as shown in Figures 7–9 and S1–S4), the average differences remained low.

Over the CMR (mountainous region enclosed in red in Figure 2), however, an increase in precipitation was found for increasing mountain height. Figure 10 shows the difference in precipitation averaged over the CMR (black line) with the 95% CI in grey shading. After TC landfall ($t > 0$), the absence of the CMR as well as the lower mountain height result in a significant decrease in post-landfall precipitation over the CMR (Figure 10a,b). Conversely, the higher mountain height corresponded to a significant increase in precipitation as early as 9 h prior to landfall (Figure 10c). However, the decrease in precipitation observed in the NTCs Imbudo and Prapiroon in the Enhanced CMR, seen as the red dashed lines in Figure 10c, reduced the overall mean precipitation (black bold line) of all TC events. This means that for the eight TCs in our study, NTCs and STCs responded differently to different orography changes. STCs tended to receive more precipitation over the CMR when mountain heights increased. On the other hand, NTCs tended to have less precipitation when mountain heights increased.

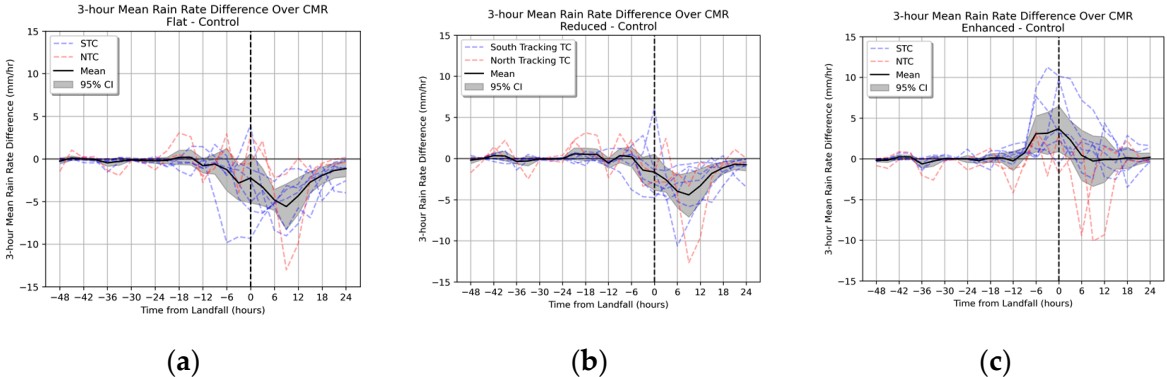

**Figure 10.** Similar to Figure 5, but for CMR precipitation difference for (**a**) Flat—Control, (**b**) Reduced—Control, and (**c**) Enhanced—Control.

### 3.4. Relationship between Precipitation, Wind Speed, and Mountain Slope

The TC tracks in the different model experiments (Figure 6), as well as the precipitation difference maps (Figures 7–9 and S1–S3) in the previous section, demonstrated the sensitivity of precipitation on TC movement. The tracks of the TCs changed as a response to the different CMR heights; thus, precipitation fell along different regions of Luzon. However, because there were no systematic patterns between the TC tracks and overland precipitation between different orography experiments, it was difficult to quantify the changes in TC precipitation over all of Luzon due to TC track changes. As mentioned in Section 3.3, using a simple and idealised statistical decomposition of TC precipitation due to the change in track, this effect is small.

However, analysing the localised interactions of winds and the slopes on CMR enabled us to evaluate the direct role of mountain slopes on precipitation, regardless of TC track. As such, in this section, we focus on precipitation and physical interactions along the slopes of the CMR itself, by investigating the empirical relationships between precipitation and individual physical variables such as the zonal winds 1 km above the surface approaching from the plains ($u_p$), zonal winds moving up the mountain slope ($u_s$), barrier height ($h$), barrier slope ($h/dx$), and moist Froude Number ($F_w$).

$F_w$ is calculated according to the following equation, as in Chu and Lin [37]:

$$F_w = \frac{u_p}{Nh} \tag{1}$$

where $u_p$ is the wind speed coming from the plains perpendicular to the barrier, $N$ is the moist Brunt–Väisälä frequency, and $h$ is the barrier height. According to Chu and Lin [37], precipitation is constrained within the plains and upslope regions of the mountain for low to moderate $F_w$ with precipitation falling near the peaks and the downslope regions (leeside) for higher $F_w$. Additionally, Sinclair [6] mentioned that precipitation is assumed to be enhanced when air is forced over a barrier—this enhancement of precipitation is proportional to the low-level flow and the barrier slope. This is especially true for wider barriers which result in more efficient water vapour removal along the mountain range [2].

We focused on three regions (Northern, Central, and Southern) towards the east of the CMR (Figure 11). These were further divided into mountain slope (a, b, and c) and plains regions (d, e, and f) inside the solid and dashed boxes in Figure 11, respectively. Three-hourly data from 24 h before until 24 h after landfall were used in calculating these variables. For all TCs, each variable was compared against the mean rain rate along the mountain slope (a, b, and c in Figure 11) for all four orography experiments (Control, Flat, Reduced, and Enhanced). This was to establish empirical relationships between winds, mountain height, and mountain precipitation.

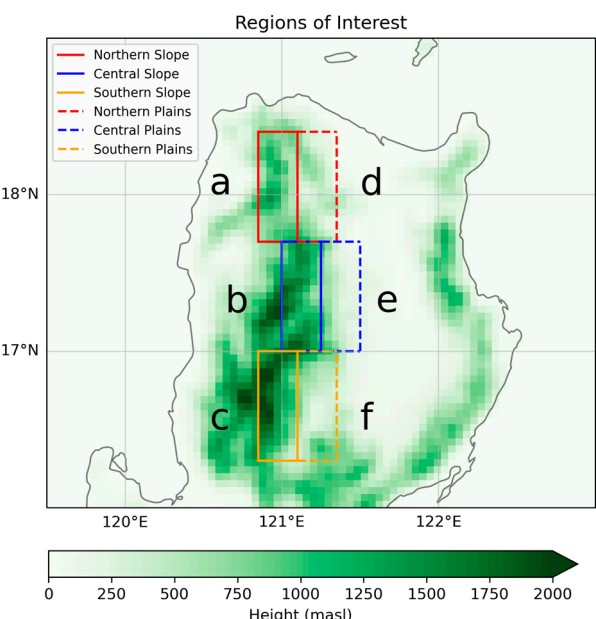

**Figure 11.** The Northern (red), Central (blue), and Southern (yellow) regions towards the east of the CMR. The solid boxes (a, b, and c) enclose the eastern slope of the mountain, while the dashed boxes (d, e, and f) show the plains regions towards the east of the mountains.

We first compared the mean precipitation rate in the eastern slope of the mountain range with the average mountain height ($h$) and incoming westward zonal wind speeds along the slope ($u_s$), excluding eastward downslope winds coming from the western side of the CMR. During the TC landfall period, there was a weak positive relationship between precipitation and perpendicular upslope wind speed $u_s$ (Figure 12a). Similarly, precipitation rate tended to be greater with taller mountains (Figure 12b). These results show that, considered on their own, neither the wind speed nor the mountain height is a good indicator of rainfall along the mountain slope.

The relationship between precipitation and wind speeds is somewhat weak and there is large variability between precipitation rates for the same mountain heights; therefore, we next turned to $F_w$, which is the ratio of wind speeds from the plains, $u_p$, and mountain height, $h$ (all divided by stability, $N$). Figure 12c shows that the relationship between precipitation rate and $F_w$ is not significant, indicating that $F_w$ is not a good descriptor of mountain rainfall either. Notably, the highest values of rainfall were found with low values of $F_w$, hinting at a negative relationship as found in the low-flow regime (Regime I) where convective cells were generated over the upslope of the mountain for stratified low-bulk flow [37].

The precipitation rate along the mountain was well correlated with the product of the upslope zonal wind and mountain slope ($u_s \times h/dx$). Figure 12d shows that $u_s \times h/dx$ is strongly and positively correlated with the precipitation rate. When the mountain becomes steeper and TC upslope winds become stronger, TC precipitation increases. This is presumably due to the combined effect of stronger winds and steeper slopes, which both favour the mechanical uplift of moist winds, causing more condensation and higher amounts of precipitation. This is consistent with the concepts of Sinclair (1994) [6] and Rostom and Lin (2021) [7], that a strong vertical motion due to mountain slopes results in the orographic enhancement of precipitation.

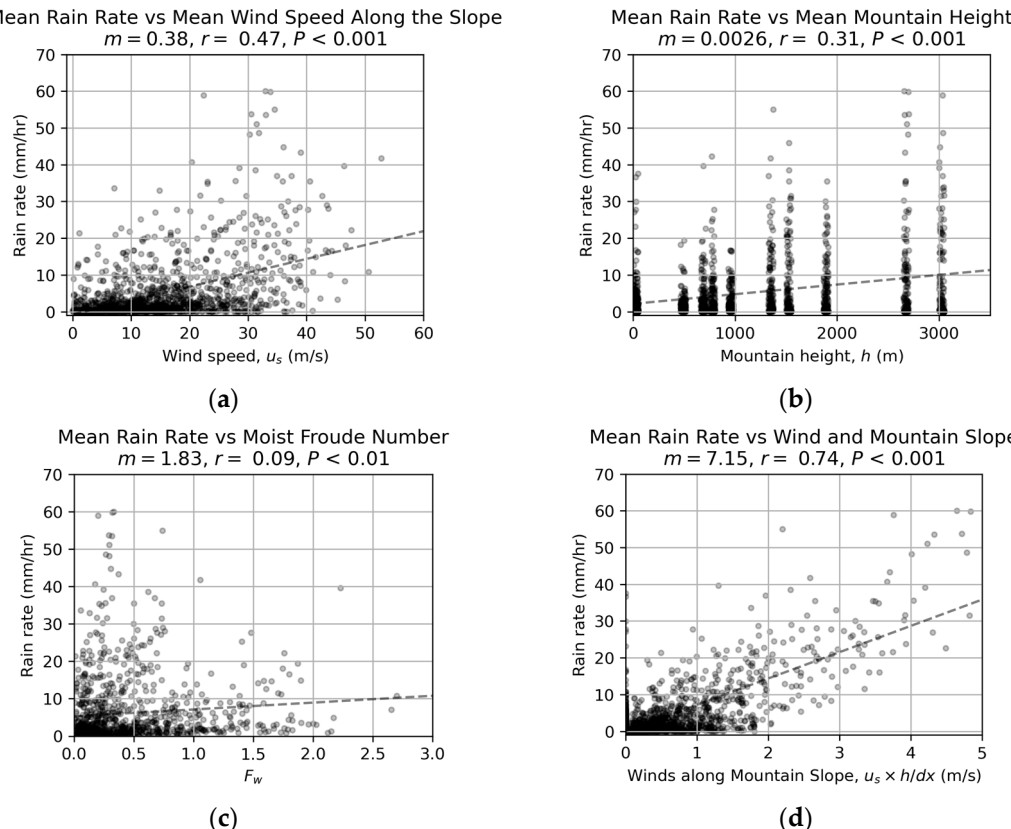

**Figure 12.** Mean rain rate over the slopes of the CMR as compared with (**a**) the upslope wind speed, $u_s$, (**b**) mountain height, $h$, (**c**) $F_w$, and (**d**) the product of wind speed, $u_s$, and mountain slope. Each data point represents 3-hourly periods at different times around TC landfall: from 24 h before landfall, the time of landfall, and up to 24 h after landfall. The comparisons that include wind speeds (**a**,**d**) do not include winds coming from the west. The dashed lines show the least-squares regression slope for each relationship. The Pearson correlation coefficient ($r$) and $p$-value ($P$) are included above each plot.

To highlight the relationship between winds, slope, and precipitation, we took snapshots of the vertical–zonal cross-section averaged across the latitude range of the Central region (blue enclosed region in Figure 11). For Koppu, a TC that showed dramatic differences in orographic impacts between the different experiments, as the product of westward wind and slope at each longitude increased, precipitation similarly increased along the mountain slopes (Figure 13). Due to the absence of the CMR in the Flat experiment, minimal precipitation was observed along this section of Luzon (Figure 13a). As the terrain heights increased in the Reduced experiment (Figure 13b), precipitation was observed along the general region of the CMR (120.4° E to 121.6° E). Steeper mountain slopes in the Control (Figure 13c) and Enhanced (Figure 13d) showed stronger vertical uplift along the slopes, as well as higher amounts of precipitation. As the TC winds flowed above the mountain top (between 121.0° E and 121.1° E), the average TC winds started to descend, and we observed smaller amounts of precipitation towards the leeward side of the mountain. These demonstrate the increase in precipitation for stronger upslope winds flowing along steeper slopes. Low amounts of precipitation were observed in the absence of the CMR; however, as the mountain slope increased from Reduced, to Control, and finally, to Enhanced, horizontal wind speed increased in addition to precipitation along the windward mountain slope.

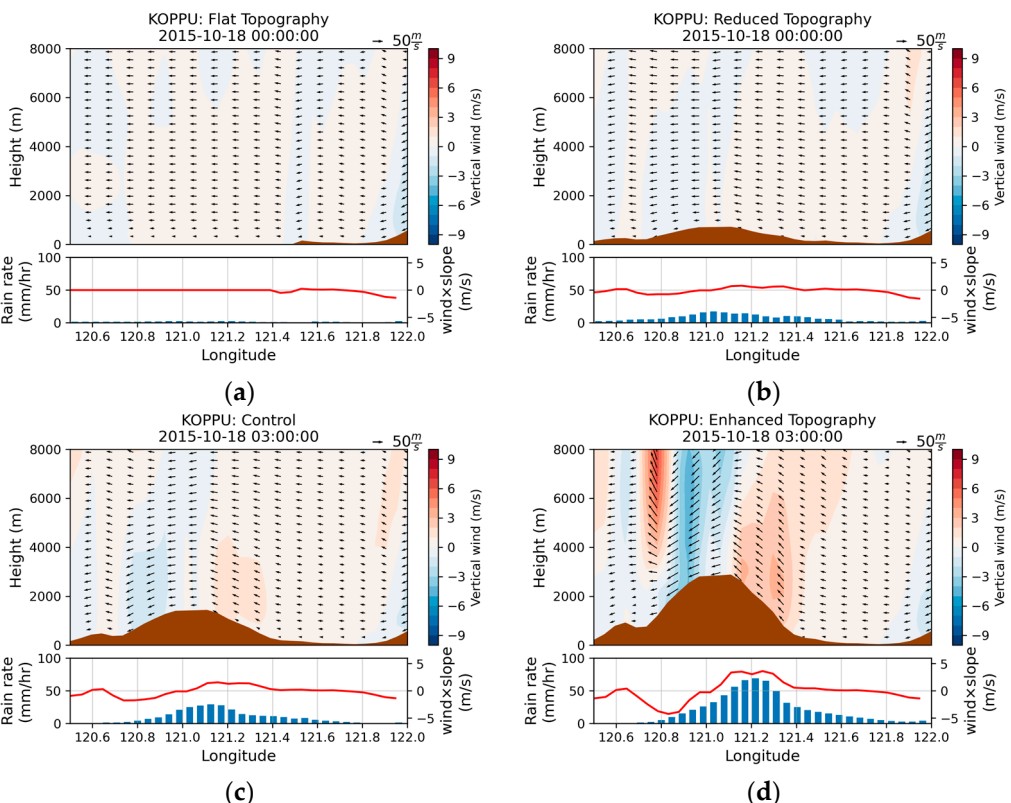

**Figure 13.** Zonal–vertical cross-section (averaged across the latitude range of the "Central" region shown in Figure 11) of vertical wind (red–blue shading) and mountain height (filled brown curve) during TC Koppu landfall (upper part of each panel) and the average precipitation along the cross-section (lower part of each panel) for (**a**) Flat, (**b**) Reduced, (**c**) Control, and (**d**) Enhanced. In the lower panels, red lines represent the product of the wind speed and slope, and blue bars show the average precipitation.

## 4. Discussion

This study was motivated by the need to understand the effects of local orography on TCs making landfall in Luzon. Although there have been studies on the effects of orography on precipitation [1–3,6–8], the effects of Luzon's orography on TC characteristics and TC precipitation are not well understood. As such, we examined the effect of the Cordillera Mountain Range, or the CMR, the largest mountain range in Luzon, on the precipitation and characteristics of TCs making landfall in the region.

Using the WRF model, we found that during TC landfall, precipitation along the mountain range slopes increased with increasing CMR height on average, whereas there were no significant changes for TC movement speeds and TC position for different terrain profiles. TCs tended to weaken for the steeper CMR in the Enhanced experiment compared with the Control. In contrast, the weakening of TCs between Flat or Reduced and Control was not apparent. These results suggest a nonlinear relationship between CMR heights and the characteristics of the eight TCs simulated in this study. Moreover, the results also suggest that the actual CMR has a minimal effect on TC intensity in these case studies, as intensity did not differ significantly between Control, Flat, and Reduced experiments. However, we still recommend that more simulations be performed for additional terrain heights and additional TCs to further evaluate the effects of terrain heights on TC intensity.

We also highlight the sensitivity of TC precipitation to the TC track—precipitation along the slopes of the CMR increased for south-tracking TCs (STCs), whereas there was less consistency for north-tracking TCs (NTCs). This is due to the direction of the swirling TC winds moving towards the western region of the CMR, shifting the windward side of the CMR from east to west, causing precipitation to fall over the South China Sea region

instead of land, and the CMR itself serving to block the eastward moisture flow. For almost all TC cases in this study, orographically induced TC rainfall varied with modification of orography. In the case of Nari and Bebinca, their tracks also shifted, particularly northward in the Enhanced experiment. Although we did not see a systematic pattern between the TC track shifts and orographic changes, TC track shifts could also have non-negligible contributions in precipitation differences in the CMR, especially for more intense TCs.

Stronger TCs making landfall in Luzon have higher chances of causing extreme precipitation [5]. As shown in the results of our study, mechanical uplifts caused by stronger winds blowing up steeper slopes results in higher amounts of precipitation along the CMR (Figures 12d and 13). Our results show that TC precipitation mainly increased towards the eastern regions of the CMR; the taller mountain ranges in the Enhanced experiment also weakened TCs (compared with Control) as they made landfall. The effects of the mountain range experiments on TC rainfall varied with TC cases, highlighting the complexity of the mountain, wind, and rainfall relationship. Although taller mountain ranges may indeed weaken TCs as they make landfall, the combination of steeper slopes and still-strong upslope winds may cause higher amounts of precipitation along the mountain slopes.

Modelling yielded clear representations of the physical processes involved; however, model errors may have affected our results. Similarly, only CMR height was varied in this study, while keeping its land use categories and the heights of the other mountain ranges of Luzon constant. The other mountain ranges may have had profound impacts on TCs as well that are yet to be understood and will be considered for future studies. The interactions of orography with other large-scale meteorological phenomena (such as monsoons or non-landfalling TCs) may also be considered for future research.

Knowing the effects of mountain ranges of Luzon on TCs can be helpful for the Philippines, and for other countries in the Northwest Pacific region. Man-chi and Chun-Wing [38] previously mentioned that TCs crossing Luzon from east to west are more likely to affect countries to the west of the Philippines (e.g., Hong Kong, Vietnam, etc.), and understanding the effects of the mountain ranges of Luzon on TC intensity may be helpful in anticipating possible hazards in those regions. For the Philippines, understanding how TCs interact with mountain ranges may help in identifying the local enhancement of precipitation due to stronger moist upslope winds across mountain ranges. However, because taller mountain ranges themselves may indeed weaken TC intensity (and hence, winds), there are further nuances to consider in terms of anticipating the amount of precipitation incoming TCs may bring.

## 5. Conclusions

In this study, we used the Weather and Research Forecast (WRF) model to investigate the effect of different heights of the Cordillera Mountain Range (CMR) in Luzon, Philippines, on tropical cyclone (TC) characteristics (such as movement speed, intensity, and wind strength) and TC precipitation. Although changing the mountain height is not possible in real-world scenarios, this study furthered understanding of the interactions between TCs and mountain ranges.

From simulations of eight TCs for four different CMR heights (Flat, Reduced, Control, and Enhanced), we found that TC intensity weakened as early as 21 h prior to landfall for a taller CMR (Enhanced). On the other hand, TC intensity is not as sensitive to lower CMR heights (Flat, Reduced, and Control). We also found that TC precipitation increased for taller mountain heights due to stronger TC winds impinging on steeper mountain slopes, causing stronger uplift in the process. The stronger ascent of moist TC winds causes higher amounts of precipitation, especially along the mountain slopes. Furthermore, we highlight the sensitivity of TC precipitation to track, as the increase in precipitation over CMR is especially true for southern tracking TCs, with the easterly TC winds impinging on the slopes of the mountain range.

Although our study offers novel insights regarding the effect of different CMR heights on selected TCs, we acknowledge that TC rainfall and intensity vary with TC cases, high-

lighting the complexity of the mountain, wind, and rainfall relationship. Furthermore, we based and adapted our WRF model configuration for different parametrisation options and model settings from the existing literature, focusing on the Philippines. For further analysis of the relationship between the CMR and TCs making landfall in Luzon, we recommend that future studies validate model accuracy for different parametrisation options, simulate additional CMR heights, and finally, include more TCs in model simulations. Additionally, we recommend that the WRF model accuracy be improved with the help of data assimilation, and then validating the model accuracy against ground-based observations.

**Supplementary Materials:** The following supporting information can be downloaded at: https: //www.mdpi.com/article/10.3390/atmos14040643/s1. Figure S1: Difference in 24 h accumulated precipitation after landfall (shading) between Flat and Control experiments but zoomed in to highlight the difference in precipitation over the CMR's terrain. Contour lines for 500 m and 1200 m heights are labelled accordingly. Figure S2: Similar to Figure S1, but for the difference between Reduced and Control experiments. Figure S3: Similar to Figure S1, but for the difference between Enhanced and Control experiments. Figure S4: Regions where differences from the Control are significant at $\alpha = 0.05$ for the (a) Flat, (b) Reduced, and (c) Enhanced experiments. Notably, only STCs are included in evaluating the significance of the Enhanced experiments.

**Author Contributions:** Conceptualisation, B.A.B.R.; methodology, B.A.B.R., C.E.H., R.K.H.S., X.F. and G.B.; software, B.A.B.R.; validation, B.A.B.R.; formal analysis, B.A.B.R.; investigation, B.A.B.R.; resources, G.B.; data curation, B.A.B.R.; writing—original draft preparation, B.A.B.R.; writing—review and editing, B.A.B.R., C.E.H., R.K.H.S., X.F. and G.B.; visualisation, B.A.B.R.; supervision, C.E.H., R.K.H.S., X.F. and G.B.; project administration, B.A.B.R. All authors have read and agreed to the published version of the manuscript.

**Funding:** This research received no external funding.

**Institutional Review Board Statement:** Not applicable.

**Informed Consent Statement:** Not applicable.

**Data Availability Statement:** Data used to create the above figures can be publicly found at https: //doi.org/10.5067/GPM/IMERGDL/DAY/06 (accessed on 5 August 2022); https://www.ncdc. noaa.gov/ibtracs/ (accessed on 22 September 2019); https://cds.climate.copernicus.eu/cdsapp#!/ dataset/reanalysis-era5-single-levels (accessed on 11 February 2021) distributed under a Copernicus Products licence. Large amounts of model outputs were created from multiple experiments; therefore, it is not possible to provide the full dataset. The model specification and methodology should enable users to recreate a similar dataset. WRF configuration files and post-processing scripts can be found in https://doi.org/10.5281/zenodo.7398219 (accessed on 5 December 2022), while modified terrestrial data for the orography experiments (Control, Reduced, and Enhanced) are provided in https://doi.org/10.5281/zenodo.7398421 (accessed on 5 December 2022).

**Acknowledgments:** The authors would like to thank the anonymous reviewers for their valuable comments. B.A.B.R. was supported by a scholarship under the Commission on Higher Education of the Philippines's Joint Development of Niche Programmes agreement with the British Council. B.A.B.R. would also like to thank Nicholas P. Klingaman and Rafaela Jane P. Delfino for discussions during different stages of this research. R.K.H.S was supported by the National Centre for Atmospheric Science. X.F. was supported by the UK Met Office Weather and Climate Science for Service Partnership for Southeast Asia, as part of the Newton Fund. This study used JASMIN, the UK's collaborative data analysis environment (https://jasmin.ac.uk (accessed on 27 October 2022) [39]). WRF post-processing was performed using xarray [40] and wrf-python [41].

**Conflicts of Interest:** The authors declare no conflict of interest.

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
