# Peer review of "The Effect of the Cordillera Mountain Range on Tropical Cyclone Rainfall in the Northern Philippines"

_atmosphere, doi:10.3390/atmos14040643_

Round 1

Reviewer 1 Report (New Reviewer)

This study unveils the Effect of the Cordillera Mountain Range on Tropical Cy-2 clone Rainfall in the Northern Philippines. Overall, this is a rigorously designed study with a significant synthesis. I like the idea of simulating the Tropical Cyclones with four different CMR orographic elevations including Control, Flat, Reduced, and Enhanced The approach is efficient and the inferences derived from the study meet the aimed scope of the study. I evaluate this as a well-written manuscript with rare places that deserve major changes. I have some gross comments and detailed suggestions that need authors to pay attention to and modify to improve the quality of the study. Thereafter, I may accept the submission for publication.

Title: Good and to the point

Abstract: Sounds good and very clear.

Introduction

The introduction is very good; the authors demonstrate a thorough knowledge of the published literature and highlight the importance and background to carry out this investigation. However, I would suggest adding a short paragraph explaining the aims and objectives of this research rather than narrating the orientation of the upcoming chapters/sections.

Materials and Methods

Consider adding a flow chart or at least a description of data sets and sources. A table of all the data used could be helpful to the reader.

Overall, the methods are well explained.

Results

The results are clear and well-presented.

Figure 3 is good.

Figure 6 is good

Figure 7 is interesting – good

Figure 17 is good too.

Overall, I liked the quality of the figures. However, it might be good to move some figures to an appendix to keep the focus on the key finding of the study in the result section.

Discussion and conclusion

Good.

Author Response

Please see the attachment for our response.

Reviewer 2 Report (New Reviewer)

Comments to the authors (if any)

I am writing in regard to manuscript ID atmosphere-2256259 entitled “The Effect of the Cordillera Mountain Range on Tropical Cyclone Rainfall in the Northern Philippines,” which was submitted to the Atmosphere MDPI journal by Racoma et al,. The subject of the paper addressed is within the scope of the journal. After careful review, I can say that the manuscript is very interesting and has scholarly importance. To improve the manuscript, I have a minor revision that I would recommend the author consider:

·         Lines 32-36, and Lines 32-36. In these paragraphs, some think are missing. Please improve them.

·         Lines 45-73. This section is too short. In addition, most of the used references are not updated. Please improve this section with new studies of the same framework.

·            The authors need to add an evaluation of the regional model at least for two events. We have to make sure that the selected parameterization does not overestimate for all the used time scales and hence performs well during tropical Cyclone events. This simulation needs to be performed

·               Discussion and conclusions must be written each one on separated section.

Author Response

Please see the attachment for our response. 

Reviewer 3 Report (New Reviewer)

Manuscript entitled “The Effect of the Cordillera Mountain Range on Tropical Cyclone Rainfall in the Northern Philippines”

Overall, this study and its results are interesting, with proper explanation. However, scientifically many aspects need to be considered. I recommend a major revision before this paper can be accepted for publication.

General comments:

1.     Line 115-119, “For the inner domain, although ………… physically-based convective rainfall processes.” the lines are unclear. Rewrite the lines with appropriate references. 

2.     Line 121-122, What is the total length of the forecast for each TC event? What is the effective lead of forecast (hours or days) for each Tropical Cyclone?

3.     Line 121-122, “To allow for model spin up within the domain, simulations were 96 hours before TC landfall and ended at 24 hours after landfall.” What is the scientific reason for 96-hour spin-up time? For me, it looks 96-hour spin-up time is too large. Also, so data assimilation has been made to initialize WRF model forecasts. Did the author verify the model hydrometer forecast accuracy after 96-hour forecast? In addition, how did the author choose the length of the forecast (120 hr)? Is it fixed for all the TC events? Is there any relation between the TC genesis and the landfall forecast length for each Tropical cyclone?

4.     Line 119-121 and Table 1. Model configuration “Adaptive time-step” need to explain the advantage of adaptive time-step as compared to static time-step. Is the forecast output significantly altering with this method?

5.     Table 1, “Microphysics = WRF Single-Moment 6-class microphysics scheme ”, Did the authors use different microphysics (Thomson or NSSL-2 moment scheme) to check the improvement of the amount of rainfall during TC landfall? For the high-resolution WRF model, choosing a cloud microphysics scheme is important. Multiple studies highlight the issue in their works. I know this is a challenging task, but acknowledging it for future work will be good.

Author Response

Please see the attachment for our response. 

Reviewer 4 Report (New Reviewer)

The article focuses on the effects of CMR on TC rainfall via WRF simulation. Several parameters are examined. Basically, this study provides useful insights for better understanding of topograhy-affected TC rainfall. However, there are some points that should be addressed. Thus, this reviewer suggest the article be publishable after minor revision.

1. Please clarify the meaning of alpha on its first apperance (Line 151 and in the abstract). Signifcance level?

2. Section 3.1: the errors of TC track (Control VS IBTrACS) seem to be not so small. Please add some explanations to better understand such errors.

3. Lines 183-184: The authors attribute the difference between Control and IMERG to the large uncertainty in measuring precipitation over mountainous terrain via IMERG. Why should not attributed it to modeling errors/uncertainty involved in the Control? Why not compare the Control results with those regarded as credible (e.g., ground-based measurements from weather stations)

4. The conclusion part is a bit lengthy. Please shorten it.

Round 2

Reviewer 3 Report (New Reviewer)

The authors have addressed my comments and concerns from the first round of review. Compared to the previous version, the manuscript is better.

Author Response

Thank you very much!

This manuscript is a resubmission of an earlier submission. The following is a list of the peer review reports and author responses from that submission.

Round 1

Author Response

Please see the attachment, which includes responses to both reviewers' comments. 

Reviewer 2 Report

I found the paper and the topic interesting. It is fairly well written as well. My major issue with the presentation has to deal with the methodology chosen to address the topic of orographic impacts on precipitation on the Luzon Cordillera Mountain Range (CMR). As noted by the authors there was no acceptable observed estimation of the precipitation, so they chose to compare the impacts of the orography through comparison of a model control simulation to simulations with modified orography. That was a fairly sound approach, except the model simulations didn’t really provide a detailed representation of the orographic impact spatially or temporally, only providing a broad brush explanation of the impact that the orography has on the precipitation distribution. Rather than using the model simulations to examine what drives the impacts, the authors usually resorted to speculation on what the drivers were, or state that they are beyond the scope of the paper. While there results seem plausible, they lack much of a clear characterization of the impacts and their causes. In my opinion the paper needs major revision before acceptance. I provide a number of suggestions and comments in the attached PDF of the manuscript that highlight particular questions and issues I noted.

Author Response

(The authors gave the same response as above.)

Round 2

Reviewer 1 Report

I have no more questions. The paper can be accepted for publication now. Meanwhile, I still recommend the authors to add more sensitivity experiments by setting a series of terrain height, maybe not in this study, but in future studies.